# Determining Current Medications Usage within a Cohort of Patients in the UK—A Descriptive Retrospective Study

**DOI:** 10.3390/healthcare10122421

**Published:** 2022-11-30

**Authors:** Adel S. Alhlayl, Haitham A. Alzghaibi, Qazi Mohammad Sajid Jamal

**Affiliations:** 1Department of Academic Directorate for Training and Research Affairs, Hail Health Cluster, Hail 55471, Saudi Arabia; 2Department of Health Informatics, College of Public Health and Health Informatics, Qassim University, Albukayriah 52741, Saudi Arabia

**Keywords:** multiple sclerosis, depression, SAIL databank, amitriptyline, anxiety

## Abstract

Swansea University’s United Kingdom (UK) Multiple Sclerosis (MS) Register is a platform that contains information on more than 17,600 people with MS living in the UK. The register has been in operation since 2011 and represents comprehensive information about people living with MS in the UK. It is considered the first register of its kind that can link information from patients to clinical data and has been established to answer different information needs about MS. **Aim**: To elucidate the trends in patterns of medicines currently used by people with MS in the UK MS register. **Methods**: This study follows an exploratory descriptive design using the UK MS register as data resource. A number of 4516 people completed the EQ-5D survey out of 8736 people who have given their consent to answer online questionnaires which represents around 52% of the register total population. Descriptive analysis and tests were performed with SPSS to address the research objectives. **Results**: There are several medicine names entered by people with MS in their profiles. These medicines are used either to manage MS symptoms or to treat its associated complications. Among the medicine types revealed in this study, disease modifying drugs (DMDs), muscle relaxants, and anticonvulsants are the medicine types mainly used by people with MS followed by antidepressant and antianxiety medicines. **Conclusions**: From the antidepressants used most widely, amitriptyline was chosen as a subject medicine for further investigation in the remaining studies of this research due to its high frequency use, the elevated depression rates discovered among people with MS who seek information on it online, and the high online content noted on websites about this medicine.

## 1. Introduction

Swansea University’s UK Multiple Sclerosis Register is a platform that contains information on more than 17,600 people with MS living in the UK [1]. The register was created in 2011 in order to represent comprehensive information about people living with MS in the UK [2]. It is considered the first register of its kind that can link information from patients to clinical data [1] and has been established to answer different information needs about MS [3]. Joining is open to any person who has a confirmed diagnosis of MS in the UK and is over 18 years old [2]. People who register there are asked to respond to a group of questions about their conditions, their therapies, and the impact MS has on their lives [2]. People with MS provide their information in the register from different places in the UK, so it can be considered a national deployment [4].

The register gathers and links electronic information from three essential resources: from clinical information systems used in different NHS hospitals and treatment centres across the UK; from normal routinely collected health data; and significantly, directly from people living with MS via a purpose-built web portal [5]. The register can anonymously link information from the triple sources at personal level while maintaining privacy [5]. As a result of unifying such information from the three different resources, the UK MS Register is pioneering in providing chances to study MS related topics in linked information [5]. The register is built on verified techniques and vigorous data control measures within the Secure Anonymised Information Linkage (SAIL) system developed by the Health Information Research Unit (HIRU) [5,6,7].

Multiple sclerosis is a chronic ailment that currently has no cure [8] but there are a variety of medicines prescribed to people with MS to overcome MS and other accompanying diseases. For example, major depression is a condition that people with MS are normally expected to experience as there is a relation assumed between MS and depression [9]. Investigations have revealed considerably high ratios of anxiety and depression among MS patients [10,11]. The life-threatening hazard of depression in people suffering from multiple sclerosis was predicted to be higher than the normal public [5,9,12,13,14].

Different studies describe the high frequency rates of anxiety and depression among persons living with MS [13,14,15]. A study by Jones, Ford [5] has sourced replies from the web portal of the UK MS Register and illustrated the depression and anxiety summaries of MS populations. The study confirmed that among 4178 respondents, anxiety and depression were remarkably high with more than 54% with anxiety and 47% with depression. This indicates that there are other medicines that could be prescribed to people with MS to manage anxiety and depression, in addition to medicines prescribed specifically for MS. The symptoms of anxiety and depression need to be treated together with the symptoms associated with multiple sclerosis. Mohr, Hart [16] discovered that out of 260 persons with MS investigated, more than 65% had not been prescribed with depression medicines while 35% had been prescribed with antidepressant medicines in various dosing administrations.

In 2008 the Food and Drug Authority (FDA) approved venlafaxine as a medication off-label in multiple sclerosis to treat neuropathic pain (dysesthesias), anxiety, and depression [17]. In 2017 FDA and the European Medicines Agency (EMA) approved the drug ocrelizumab for the treatment of early primary progressive multiple sclerosis [18]. In 2019 the FDA approved another drug Mavenclad (cladribine) as a tablet to treat relapsing forms of MS in adults [19]. A previous study explored the antidepressant medication role in MS individuals and their investigation revealed that drug compounds from the tricyclic class significantly showed beneficial mechanisms of clomipramine, desipramine, trimipramine, imipramine, and doxepin in the treatment of MS [20,21]

## 2. Methods

### 2.1. Aim and Objectives

#### 2.1.1. Study Aim

The aim of this study is to elucidate the trends in patterns of medicines currently used by people with MS in the UK MS register.

#### 2.1.2. Objectives

To investigate records of people with MS participating in the register and determine the rates of anxiety and depression incidences.To analyze HADS and EQ-5D web portal questionnaire results that describe the lifestyles of people with MS relating to anxiety and depression disorders.To analyze medicinal records of people with MS and find medicines that are frequently used by them to manage MS and its concurrent ailments.

### 2.2. Design

This study follows an exploratory descriptive design using the UK MS register as data resource. To achieve the study’s aim and objectives specified above, the health status of the participants was analysed, the anxiety and depression rates were calculated, and the patterns of medicine usage were obtained using the portal built in questionnaire outcomes.

### 2.3. Population and Sample

There are more than 8736 people with MS available in the register who have given their consent to answer online questionnaires [22]. This represents the population used in this study. However, not all members of the register were expected to answer the web portal built in questionnaires. According to Jones, Ford [3] only 4516 people completed the EQ-5D survey out of 8736 people with MS registered in the portal at that time which represents around 52% of the register total population. Although this study may have a larger population size than the Jones, Ford [3] study, it is expected that the sample size will be lower than the maximum possible.

### 2.4. Data Collection Instruments

The UK MS Register represents a flexible platform for the delivery of a range of information about people with MS in the UK including demographics, diagnosis, type of MS, and medicine records [4]. The web portal hosts a diversity of validated questionnaires prepared by researchers to achieve a complete picture of the health status and welfare of people with MS [4,5]. These validated questionnaires include the hospital anxiety and depression scale (HADS), health-related quality of life measure (EQ-5D), and medicine records [4,23,24,25,26].

The HADS questionnaire measures anxiety and depression using seven questions for anxiety and seven questions for depression [27]. For both anxiety and depression scales, Snaith [26] categorised HADS anxiety and depression scores as follows: (0–7) = normal, (8–10) = borderline/abnormal, and (11–21) = abnormal.

The EQ-5D questionnaire contains two parts; health status description and evaluation [28]. In the description part, health status is assessed in terms of five dimensions (5D); mobility, self-care, usual activities, pain/discomfort, and anxiety/depression. Each dimension of the description part is measured by a three-level scale; having no problems, having some or moderate problems, being unable to do/having extreme problems [28]. The participants can choose which best describes their health status and the chosen level that can be coded as a number 1, 2, or 3. Thus a participant’s health status can be defined by a 5-digit number ranging from 11,111 (having no problems in all dimensions) to 33,333 (having extreme problems in all dimensions). In the evaluation part, the respondents can evaluate their overall health status using a visual analogue scale (EQ-VAS).

### 2.5. Ethical Consideration

The UK MS Register as a research database has ethical permission from the South West Central Bristol National Research Ethics Committee 11/SW/0160 to ask participants about their currently used medications [5]. Under this consent of ethics, information collected from NHS clinical systems together with data from the normal administration resources are anonymised and linked in the portal. This procedure can be done using the SAIL methodologies only if agreements to the portal terms of service and written informed participant consents have been gained in clinics [5]. The register information could be available for analysis by outside investigators who can meet the controlling and governing prerequisites [5]. In this study, the researcher has met the required prerequisites as shown below in addition to the research ethical approval by Swansea University Medical School Research Ethics Committee (reference number: 120716-536264@swansea.ac.uk).

### 2.6. Procedure

In order to get access to the UK MS Register data, the researcher needs to complete an information governance programme on the topic of safe researcher training. Moreover, the researcher had to sign a research agreement with the portal working team. The register holds only anonymous information with only user ID as population member identification. However, medicine names entered are not arranged in any sort of information structure and it is left to participants to type in their medicine names.

### 2.7. Data Analysis

The data relating to the user IDs could be exported from the register database as excel files and must remain inside secure research environment (i.e., data provided was found listed in the register database as tables against user ids). Using the Microsoft Excel Query Feature, the data was joined from two exported Excel worksheets in a new workbook to create a new database based on the common filed user id. The joint data in the new excel workbook was double checked by randomly comparing several rows of the same user id against the original database. Descriptive analysis and tests were performed with SPSS (v.25) to address the research objectives. Descriptive analysis was conducted to find demographics and background information, anxiety and depression incidences, and patterns of medicines used by the study population. The chi-square test was performed to test for associations between different categorical variables. In addition, logistic regression analysis was carried out to determine if the obtained results of EQ-5D and HADS questionnaires predicted the use of some selected medicines.

Regression analysis can be used to investigate relationships between study findings and their independent variables “predictors”, so that different outcomes can be forecast [29,30,31]. In binary incidents, e.g., mortality yes or no incidents, logistic regression is recommended to analyse the influence of different predictors on a binary result by quantifying the unique contribution of each predictor after controlling the others [29]. Logistic regression can identify—using logit scale—notable linear combinations of predictors that have a significant probability in influencing the observed findings [29].

## 3. Results

### 3.1. Demographic and Background Information

The total number of respondents was 5940 which represents about 56% of the study population. This rate is nearly close to the percentage obtained by Ford, Jones [4] and the above response rate obtained by Jones, Ford [3] taking into consideration that the Ford et al. (2012) data were collected immediately after launching the portal.

More information was collected about participants’ **gender**, 73% were women, 26% were men (4336 women: 1545 men), and 59 participants did not record their gender (N = 5940). The **age** of respondents diverged between three categories as follows. 832 participants (14%) are in the age group of (21–40) years old. 3441 participants (57.9%) are in the age group of (41–60) years old, 1663 (28%) are above 61 years old while 4 participants did not provided information about their age (missing). The mean age of the respondents was 53.5 years with a median of 54 years and a mode of 57 years (SD = 11.5).

Among the respondents, the reported **types of MS** were as follows: 13.8% primary progressive (PPMS), 66.6% relapsing-remitting (RRMS), 7.5% secondary progressive (SPMS), and 11.7% did not know their type of MS (DKMS) while there were 24 (0.4%) missing cases (N = 5940). Furthermore, the majority of respondent had stated their **country** of origin: 4443 respondents (74.8%) stated that they lived in England, 208 (3.5%) in Northern Ireland, 630 (10.6%) in Scotland, 612 (10.3%) in Wales, while 47 (0.8%) were of unknown location (n = 5940). See Table 1 below.

### 3.2. Anxiety and Depression Incidences

Anxiety and depression incidences of people with MS were computed using the outcomes of the HADS questionnaire and the EQ-5D descriptive system (anxiety/depression dimension). It appears from Table 2 and below that almost half of the respondents (47.5%) suffer from moderate problems of anxiety and depression in the EQ-5D descriptive system while those who suffer from extreme problems are a minority (7%).

Figure 1 describes the proportions of anxiety and depression cases among participants according to the HADS score categories of Snaith, (2003): normal, borderline, and abnormal. According to this categorisation, the majority of people in the register fall into the abnormal category of anxiety and depression. This can be detailed as 50.4% of the study population being in abnormal anxiety status with 48.3% in abnormal depression status. However, about one third of the study population are in borderline of anxiety and depression.

HADS anxiety and depression total scores of the respondents were explored using descriptive statistics. The mean total anxiety score was 10.8 (SD = 3.4) with specific readings of 10.9 (SD = 3.4) for female and 10.3 (SD = 3.4) for male. The mean total depression score was 10.5 (SD = 3.1) with caption specific of 10.3 (SD = 3.1) for female and 10.8 (SD = 3.2) for male.

### 3.3. Relationship between Categorical Variables

In order to understand the association between the demographic variables against anxiety or depression, the contingency chi-square test was used to assess goodness of fit between categorical variables. An example of the data structure is shown in Table 3 for country of origin and the HADS scale. No significant association was found between the country of origin and the anxiety or depression categories of the HADS scale (χ2(6) = 8.87 *p* = 0.181; χ2(6) = 10.35 *p* = 0.111 respectively), nor between the anxiety/depression dimension of the EQ-5D Survey (χ2(6) = 10.36 *p* = 0.11). This indicates that the different anxiety or depression categories do not differ significantly in frequency among the different countries of origin.

Chi-square tests were conducted to measure the association between **gender** (males, females) and anxiety and depression. Anxiety and depression cases in the HADS scale were significantly associated with gender (Table 4, χ2(2) = 34.25 *p* = 0.000; χ2(2) = 12.86 *p* = 0.002, respectively). When examining the female sample, it was observed that 52.3% of females was under the abnormal category of anxiety compared to 45.1% of males. Similar frequencies were found in the borderline/abnormal category; however, in the normal category more males (23.6%) were present compared to females (17.9%). As for depression, on observing the abnormal category, it was obvious that males (51.7%) were more than females (47%); a difference is also observed in the normal category where females (20%) were higher than males (16.8%). For EQ-5D anxiety/depression, no significant association was found with gender, chi-squared, χ2(2) = 2.92 *p* = 0.232. The frequencies (%) of both males and females under each of the categories were similar, showing no obvious differences.

**Age** was found also to have a significant association with the anxiety and depression series of the HADS scale (χ2(4) = 146.68 *p* = 0.000; χ2(4) = 104.21 *p* = 0.000, respectively). The anxiety/depression dimension of the EQ-5D had exactly the same results as the HADS scale (χ2(4) = 45.1 *p* = 0.000). For anxiety, by observing the abnormal category it was clear that with age categories those between 21–40 years had the highest frequency (60%), followed by 41–60 years (53%), and finally those above 61 years of age (40%). However, the pattern is different in depression where the middle group showed the highest frequency (51.6%), followed by those 61 years and above (46.3%), and finally the 21–40 year group (39.2%). No differences are observed in the EQ-5D anxiety and depression across categories and age groups (Table 5).

The **type of MS** and its association with the disorders was checked. The types of MS have a significant association with anxiety and depression; HADS scale (χ2(6) = 56.57 *p* = 0.000; χ2(6) = 90.66 *p* = 0.000, respectively). The anxiety/depression dimension of the EQ-5D scale has the same results but they are insignificant (chi-square, χ2(6) = 7.82, *p* = 0.251). See Table 6 for more details. In Anxiety, by observing the abnormal category and the percentages within each of the MS types, the highest percentage was for those under types RRMS and SPMS (52.3% for both) followed by DKMS (48.1%) and PPMS (42.2%). In Depression, the highest percentage was within SPMS (61%) followed by PPMS (54.4%); both RRMS (46.1%) and DKMS (46.6%) showed similar frequencies. For the EQ-5D no differences were observed across the different MS types.

### 3.4. Patterns of Medicines Used

The data was gathered from the UK MS register over a two-month period of time. Records that contain medicines or food supplement names are included. However, duplicated records are excluded. Only medicine names with frequency of 1% or more were considered. The data was tabulated and analysed in SPSS (v.25). Descriptive statistic was applied to calculate the most frequent medicine names used by individuals on the UKMS register. Table 7 shows medicine names in categories that were found to be used prevalently by people with MS in the UKMS register over a two-month period of time. The total percentages in Table 3 are not close to 100% because there are many medicine names with very small percentage values which were excluded from Table 3.

Although there is a wide range of medicine names available in the database, it is clear from the table that there were 1195 participants (about 20.2% of the study population) not using medicine treatment. However, four medicine names categorised as disease-modifying agents (DMAs) were found to be used by people with MS in rates ranging from 1.4% for Avonex to 4.2% for Tysabri. A muscle relaxant called Baclofen was found to be used by 3.4% of people with MS according to the register records. The antidepressants Amitriptyline and citalopram were found to be used by 2.2% and 1% people with MS respectively while an antianxiety drug called Pregabalin was found to be used by 2.1% of the population. The anticonvulsant gabapentin and the painkiller paracetamol were found to be used by 2.8% and 1.1% respectively while the immuno-suppressant Tecfidera, the antiparkinsonian amantadine, and vitamin-D were found to be used by 2.6%, 1%, and 1.9% of the study population respectively

### 3.5. Logistic Regression

Amitriptyline is used in treatment of many diseases including depression [32]. Moreover, there is evidence that amitriptyline is more efficient in depression management than other antidepressants [33]. So, it is called “the gold-standard antidepressant” [34]. However, amitriptyline is taken with other medicines for pain control including neuropathic pain [35] commonly for people with multiple sclerosis [36]. Thus, assuming that the frequency rate calculated in Table 3. is for amitriptyline use in multiple sclerosis for neuropathic pain management and some mild cases of depression, it is worth determining if clinical levels (i.e., moderately/extremely cases) of anxiety and depression predict the use of amitriptyline over other options. For binary logistic regression, amitriptyline is considered a dependent variable (DV) while the outcomes of the HADS questionnaire and the EQ-5D descriptive system (anxiety/depression dimension) will be used as predictors (independent variables).

Logistic regression analysis was conducted to explain the relationship between amitriptyline use vs. other antidepressants, or amitriptyline use vs. anti-anxiety or other medicines, or amitriptyline use vs. no medicine use as binary dependent variables (DV). The EQ-5D results (anxiety/depression dimension), HADS results (anxiety cases), and HADS results (depression cases) were used as independent variables. Each of these independent variables was categorised as 1 = normal range or 2 = clinical range to make both the independent variables and the dependent variables binary in their nature. This allows prediction of a variable with a binary outcome, using other binary predictors. Table 8 below illustrates the statistically significant outcomes gained from this analysis with associated regression coefficients (B).

Most of the analyses were not significant. However, only three significant results were obtained. EQ-5D was a significant predictor of the binary outcome amitriptyline (coded 1) vs. other antidepressants (coded 2) (B = 0.886, *p* = 0.002). The positive B value indicates that those in the clinical range (coded 2) were more likely to be provided with other antidepressants (coded 2) compared to amitriptyline.

HADS depression was also found to be a significant predictor of the binary outcome amitriptyline (coded 1) vs. anti-anxiety (coded 2) (B = 0.804, *p* = 0.006). The positive B value indicates that those in the clinical range (coded 2) were more likely to be provided with an anti-anxiety medicine (coded 2) compared to amitriptyline.

EQ-5D was again found as a significant predictor of the binary outcome amitriptyline (coded 1) vs. no medicines (coded 2) (B= −0.515, *p* = 0.026). The negative B value indicates here that patients who fall in the clinical range of EQ-5D (coded 2) were more likely to be given amitriptyline (coded 1) compared to no medicines (coded 2).

## 4. Discussion

This retrospective study provides the first insight analysis into the patterns of medicines used by people with MS in the UK. While diverse proportions of medicines consumed by people with MS were found, some obvious differences were observed initially in the population characteristics. The study investigated records of 5940 people with MS who had signed up in the UK MS register and answered the web portal built in questionnaires. The portal built in questionnaires aimed to provide information about a variety of topics such as people with MS lifestyles, welfare; quality of life; and medicine details [4]. Only HADS and EQ-5D questionnaire results together with the records of medicines used by people with MS who had joined the register were considered in this study. However, demographic data about people with MS such as age, gender, date of diagnosis, and type of MS were collected during the register sign up process [5].

Over half of the study participants were found to be suffering from anxiety and depression as obtained from the EQ-5D scale outcomes. Furthermore, the majority of the study participants were found to be suffering from either abnormal anxiety and depression (50.4%, 48.3% respectively) or borderline cases of anxiety and depression (32.2%, 32.6% respectively) as described by the HADS questionnaire (Figure 1). This finding is in line with the results of other study findings such as Jones, Ford [5], Boeschoten, Braamse [15], Feinstein, Magalhaes [37], Brown, Valpiani [38], and Turner and Alschuler [39], Solaro, Gamberini [40].

Moreover, Feinstein, Magalhaes [37] stated that depression is more common in people with MS than it is in the normal population. Similarly, in this study the mean anxiety score was (10.8 SD 3.4) while the mean depression score was (10.5 SD 3.1) for the total study participants. In an earlier study for a reference group of the population in the UK it was found that the anxiety mean value was (6.14 SD 3.8) while the depression mean value was (3.68 SD 3.1) [4,41]. Thus, the results of the current study support the view that depression occurs at a higher level in people with MS than in the reference group for the population as a whole.

This study showed a significant association between gender and anxiety and depression. Females were more likely to reach the abnormal category in anxiety compared to males (52.3% to 45.10%). However, the males were more likely to be in the abnormal category in depression (51.7%) compared to females (47%). No differences were observed in EQ-5D. It was also found that the anxiety mean score was higher in women than men (HADS 10.9F:10.3M) whereas the depression mean score was higher in men than women (HADS 10.3F: 10.8M). This confirms Jones, Ford [5] findings that men were frequently more depressed than women, but women were more frequently anxious than men among people with MS in the UK MS register.

About 57% of anxiety and depression incidents were found among people whose age ranged from 41 to 60 years in this study. However, 29% of cases were found among people who were above 61 years old while only 14% of cases were found among people whose age ranged from 21 to 40 years. These results may correspond with some of the findings of Jones, Ford [5] that there is significant difference in HADS depression and anxiety scores between age groups. However, Jones, Ford [5] found that the age group of 15 to 24 years old had the highest mean rank of anxiety and the lowest mean rank of depression. The present study found that the lowest incidents of both anxiety and depression were among the age group 21 to 40 years while the highest incidents were among the age group 41 to 60 years (Table 5). Differences between the present study and that of Jones, Ford [5] may relate to the differences in age grouping.

On the other hand, anxiety and depression cases were found more frequently in this study among people with RRMS (66.8%) compared to people with PPMS (13.9%) and people with SPMS (7.6%). In more detail, abnormal anxiety and depression cases are the most among people with RRMS (69% and 64% respectively) while they are the least among people with SPMS (7% and 8% respectively). These findings confirm the findings of [5] that anxiety was most frequent among people with RRMS (56.5%) in the UK MS register, but contradict the other finding that depression was most frequent among people with SPMS (56.9%). Nevertheless, the present study adds more information suggesting that not only are the borderline and abnormal cases of anxiety and depression most frequent among RRMS but so are the normal cases. This might be because people with RRMS are about 66.6% of the UK MS register population whereas people with SPMS are about 7.5% of the population.

### Patterns of Medicines Used

Since multiple sclerosis is a disease that has no foreseen cure [8], many people with MS in this study (20.2%) were found to be either with no medicine usage or prescribed with some symptom relief medicines. Disease-modifying agents (DMAs) or DMDs (also called immunomodulator agents) are not a cure for MS, but they reduce the number and severity of MS relapses [42].

From a single point of view, it might be thought that the prevalent patterns of medicines prescribed for people with MS would be DMDs. However, although DMDs are commonly prescribed for people with MS, it is possible for doctors to treat the disease symptoms with other medicines [43]. DMDs were used in this study in frequency rates ranging from 1.4% to 4.2% as follows: Avonex 1.4%, Copaxone 1.7%, Rebif 2.2%, and Tysabri 4.2%. It is clear from these percentage values that the identified DMDs are not being prescribed to many people with MS participating in the register. This finding agrees with the conclusion of Marriott, Mamdani [44] in an Ontario-Canadian cross-sectional study that the majority of DMDs are only prescribed by a small subset of neurologists.

While DMDs are used mainly to treat neuronal degradation in people with MS, symptoms such as pain, spasticity, fatigue, bladder dysfunction, and depression are recommended to be managed and controlled as early as possible to provide optimal results for individuals and to prevent cycles of symptoms from developing [45]. Thus, a centrally acting skeletal muscle relaxant called baclofen has been used by 3.4% of people with MS in the register while an anticonvulsant called gabapentin was found to be used by 2.8%. It is common that people with MS who suffer from a primary pain undergo a secondary muscle spasm [46]. Consequently, prescribers may include these muscle relaxant and anticonvulsant medicines in prescribing to manage such spasm. Baclofen and gabapentin in combination were associated with lower rates of periodic limb movements during sleep in people with MS [47].

In addition, this study found that many types of medicines were used by people with MS in the UK MS register to manage MS symptoms. For instance, vitamin D (a vitamin or food supplement) and paracetamol (painkiller) are prescribed by a percentage of 1.9% and 1.1% respectively. This is because neuropathic pain is recognized by its relationship to multiple sclerosis [48], and therefore, painkillers with other complements are used in direct pain management [49].

Antidepressants and antianxiety medicines were also found used by this cohort of people with MS as follows. Amitriptyline was found used by 2.2%, pregabalin by 2.1%, then citalopram by 1% of the study population. These findings may signify a fact agreed unanimously by Jones, Ford [5], Michalski, Liebig [10], da Silva, Vilhena [11], Arnett, Barwick [12], Hanna and Strober [14] that depression is common and has a remarkably high incidence among people with MS and a large number of them use antidepressant medicines. Anxiety and depression in people with MS were found by de Jong and Uitdehaag [50] to coincide with less adherence to treatment and other complications such as fatigue, pain, alcohol abuse, and elevated risk of relapses.

It has been detailed that people suffering from anxiety (i.e., the normal psychiatric patients) are 3–5 times more likely to visit doctors in primary care or specialised clinics for consultations [51,52]. The internet was found to be commonly used by people with depression for medicine fact seeking [52]. Antidepressant information is initiated and disseminated online more quickly than normal published literature while the accurate scientific facts about these medicines are limited in webpages as for emotive and promotional purposes [53].

The survey study outcomes contributed much information about people with MS characteristics in the UK, their internet search behaviour, medicine records summary, favourable indicators of information quality they want to see in the web about the medicines they use, and how they prefer to see quality assessment results of such information. This study confirmed the fact reported by many previous studies that depression and anxiety are common among people with MS, while these ailments symbolizing MS are threatening and should be treated as priority in combination with the MS treatment courses. It is worth investigators in this field or area of research having good comprehension about the statistics of medicine consumption among this cohort of patients. The information analysed in this work from the secondary data provided by the UKMS register databases proved the value of such a platform for research purposes. It is valuable to investigate the health problems associated with people with MS as this topic of research is vital for clinical practice improvement and treating management of MS

## 5. Conclusions

The UK MS register as a questionnaire delivery platform offers much information about people with MS in the UK. It represents a rich source of anonymised (secondary) data that can provide extensive information for research on people with MS in the UK. The information gathered in this study confirmed other study findings relating to depression rates among people with MS. There are several medicine names entered by people with MS in their profiles. These medicines are used either to manage MS symptoms or to treat its associated complications. Among the medicine types revealed in this study, DMDs, muscle relaxants, and anticonvulsants are the medicine types mainly used by people with MS over a two-month period of time followed by antidepressant and antianxiety medicines.

## Figures and Tables

**Figure 1 healthcare-10-02421-f001:**
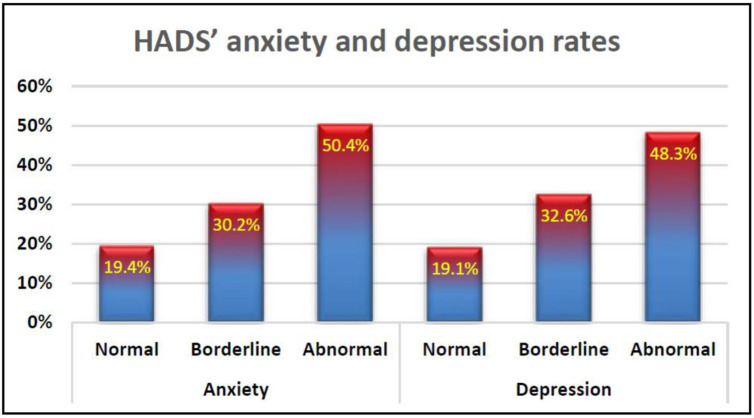
HADS anxiety and depression rates among participants from the UK MS register.

**Table 1 healthcare-10-02421-t001:** Statistics of demographic of the UK MS register population.

**Gender**		
**Response**	**N**	**%**
Male	1545	26
Female	4336	73
Overall	5881	99
Missing	59	1
Total	5940	100
**Age**		
**Response**	**N**	**%**
21–40	832	14
41–60	3441	57.9
60+	1663	28
Overall	5936	99.9
Missing	4	0.1
Total	5940	100
**Type of MS**		
**Response**	**N**	**%**
(PPMS)	820	13.8
(SPMS)	445	7.5
(RRMS)	3956	66.6
(DKMS)	695	11.7
Overall	5916	99.6
Missing	24	0.4
Total	5940	100
**Country**		
**Response**	**N**	**%**
England	4443	74.8
Northern Ireland	208	3.5
Scotland	630	10.6
Wales	612	10.3
Overall	5893	99.2
Missing	47	0.8
Total	5940	100

**Table 2 healthcare-10-02421-t002:** Statistics of EQ-5D descriptive system (anxiety/depression dimension).

EQ-5D Anxiety/Depression Dimension		
**Responses**	**N**	**%**
No problems	2643	44.5
Moderate problems	2821	47.5
Extreme problems	415	7
Missing	61	1
Total	5940	100

**Table 3 healthcare-10-02421-t003:** Statistics showing anxiety and depression frequencies according to the four countries in the UK.

Countries		HADS Anxiety			HADS Depression			EQ-5D Anxiety/Depression		
		Normal	Borderline/ Abnormal	Abnormal	Normal	Borderline/ Abnormal	Abnormal	No Problems	Moderate Problems	Extreme Problems
**England**	% within Country	19.50%	30.70%	49.70%	19.40%	32.60%	48.00%	45.40%	47.40%	7.10%
	% within Disorder	76.30%	76.80%	74.60%	76.90%	75.70%	75.10%	76.30%	74.70%	76.80%
**Northern Ireland**	% within Country	15.00%	25.90%	59.10%	20.90%	25.50%	53.60%	36.20%	55.50%	8.30%
	% within Disorder	2.70%	3.00%	4.10%	3.80%	2.70%	3.90%	2.80%	4.10%	4.10%
**Scotland**	% within Country	19.20%	30.10%	50.70%	17.70%	35.70%	46.60%	46.60%	46.60%	6.70%
	% within Disorder	10.50%	10.50%	10.60%	9.80%	11.60%	10.20%	11.00%	10.30%	10.10%
**Wales**	% within Country	19.70%	28.40%	51.90%	17.70%	31.50%	50.90%	43.10%	50.80%	6.10%
	% within Disorder	10.50%	9.70%	10.60%	9.50%	10.00%	10.80%	9.90%	10.90%	9.00%
**Total**	% within Country	19.40%	30.30%	50.40%	19.10%	32.60%	48.30%	45.00%	48.00%	7.00%
	% within Disorder	100.00%	100.00%	100.00%	100.00%	100.00%	100.00%	100.00%	100.00%	100.00%

**Table 4 healthcare-10-02421-t004:** Statistics showing anxiety and depression frequencies according to gender.

Gender		HADS Anxiety			HADS Depression		
		Normal	Borderline/ Abnormal	Abnormal	Normal	Borderline/ Abnormal	Abnormal
**Female**	% within Gender	17.90%	29.80%	52.30%	20.00%	33.00%	47.00%
	% within Disorder	67.60%	72.40%	76.20%	76.70%	74.20%	71.50%
**Male**	% within Gender	23.60%	31.30%	45.10%	16.80%	31.50%	51.70%
	% within Disorder	32.40%	27.60%	23.80%	23.30%	25.80%	28.50%
		**EQ-5D Anxiety/Depression**					
		No Problems	Moderate Problems	Extreme Problems			
**Female**	% within Gender	44.50%	48.20%	7.30%			
	% within Disorder	72.50%	73.70%	76.10%			
**Male**	% within Gender	46.40%	47.30%	6.30%			
	% within Disorder	27.50%	26.30%	23.90%			

**Table 5 healthcare-10-02421-t005:** Statistics showing anxiety and depression frequencies according to age categories.

Age		HADS Anxiety			HADS Depression			EQ-5D Anxiety/Depression		
		Normal	Borderline/ Abnormal	Abnormal	Normal	Borderline/ Abnormal	Abnormal	No Problems	Moderate Problems	Extreme Problems
**21–40 Years**	% within Age	12.80%	27.20%	60.00%	30.00%	30.80%	39.20%	43.40%	46.60%	10.00%
	% within Disorder	9.30%	12.60%	16.70%	21.90%	13.20%	11.40%	13.50%	13.60%	19.80%
**41–60 Years**	% within Age	17.00%	30.10%	53.00%	17.70%	30.70%	51.60%	43.30%	49.00%	7.80%
	% within Disorder	49.70%	56.30%	59.60%	52.50%	53.40%	60.50%	54.60%	58.00%	62.60%
**≥61 Years**	% within Age	27.10%	32.10%	40.80%	16.70%	37.10%	46.30%	49.10%	46.60%	4.20%
	% within Disorder	41.10%	31.10%	23.80%	25.60%	33.40%	28.10%	31.90%	28.40%	17.50%
**Total**	% within Age	19.40%	30.20%	50.40%	19.10%	32.60%	48.30%	45.00%	48.00%	7.00%
	% within Disorder	100.00%	100.00%	100.00%	100.00%	100.00%	100.00%	100.00%	100.00%	100.00%

**Table 6 healthcare-10-02421-t006:** Statistics showing anxiety and depression frequencies according to type of MS.

Type		HADS Anxiety			HADS Depression			EQ-5D Anxiety/Depression		
		Normal	Borderline/ Abnormal	Abnormal	Normal	Borderline/ Abnormal	Abnormal	No Problems	Moderate Problems	Extreme Problems
**PPMS**	% within Type	26.4%	31.4%	42.2%	12.7%	32.9%	54.4%	47.2%	46.9%	5.9%
	% within Disorder	18.9%	14.5%	11.6%	9.3%	14.1%	15.6%	14.6%	13.6%	11.6%
**RRMS**	% within Type	17.2%	30.5%	52.3%	21.8%	32.1%	46.1%	44.4%	48.1%	7.4%
	% within Disorder	59.1%	67.6%	69.4%	76.3%	66.1%	63.6%	66.1%	67.0%	70.5%
**SPMS**	% within Type	20.7%	27.0%	52.3%	9.3%	29.7%	61.0%	42.0%	51.8%	6.2%
	% within Disorder	8.1%	6.8%	7.8%	3.7%	6.9%	9.5%	7.1%	8.2%	6.6%
**DKMS**	% within Type	23.0%	28.9%	48.1%	17.6%	35.8%	46.6%	46.8%	46.4%	6.7%
	% within Disorder	13.9%	11.2%	11.2%	10.8%	12.9%	11.3%	12.2%	11.3%	11.2%
**Total**	% within Type	19.4%	30.2%	50.4%	19.1%	32.5%	48.4%	44.9%	48.1%	7.1%
	% within Disorder	100.0%	100.0%	100.0%	100.0%	100.0%	100.0%	100.0%	100.0%	100.0%

**Table 7 healthcare-10-02421-t007:** Most frequent medicine names used by people with MS in the UKMS register over a two-month period of time.

Therapeutic Class	Medicine Name	Frequency	Percent
None	No medicines used	1195	20.2
Disease-modifying agent	Tysabri	250	4.2
Muscle Relaxants	Baclofen	202	3.4
Anticonvulsant	Gabapentin	166	2.8
Immunosuppressants	Tecfidera	155	2.6
Antidepressants	Amitriptyline	131	2.2
Disease-modifying agent	Rebif	131	2.2
Antianxiety agent	Pregabalin	125	2.1
Multivitamins	Vitamin D	113	1.9
Disease-modifying agent	Copaxone	101	1.7
Disease-modifying agent	Avonex	83	1.4
Painkillers	Paracetamol	65	1.1
Antidepressants	Citalopram	60	1
Antiparkinsonian	Amantadine	59	1

**Table 8 healthcare-10-02421-t008:** Logistic regression analysis outcomes.

Dependent Variables	Values	Independent Variables		
		EQ-5D Anxiety/Depression	HADS Anxiety Cases	HADS Depression Cases
1. Amitriptyline vs. antidepressants (1,2)	Sig.	**0.002**	0.390	0.966
	B	**0.886**	0.314	−0.14
2. Amitriptyline vs. anti-anxiety (1,2)	Sig.	0.389	0.349	**0.006**
	B	−0.219	0.303	**0.804**
3. Amitriptyline vs. other medicines (1,2)	Sig.	0.345	0.907	0.146
	B	−0.207	0.032	0.350
4. Amitriptyline vs. no medicines (1,2)	Sig.	**0.026**	0.729	0.833
	B	**−0.515**	0.096	−0.052

## Data Availability

The author signed the data access agreement to have access to the Secure eResearch Platform. Therefore, restrictions apply to the availability of these data. Data was obtained from UK MS register and are available from the authors with the permission of the UK MS register.

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
