# Peer review of "Determining Current Medications Usage within a Cohort of Patients in the UK—A Descriptive Retrospective Study"

_healthcare, 2022, doi:10.3390/healthcare10122421_

Round 1

Reviewer 1 Report

• Antidepressants and antianxiety Medication appears several times in the results and discussion but nothing mentioned in the introduction. The authors need to add some information in the introduction to reflect them.

• What this study adds and recommendations to future work not included.

• Last paragraph in the discussion has details not belong to this study.

• Objective number 4 has not been achieved in this study. Authors need to consider whether remove it or address it in this study.

• For interpretation of the results it would be good if the reader had access to the different questionnaires such as HADS and EQ-5 to be able to see the individual questions. Could they be put as an appendix.

Author Response

Reviewer 1:

Comments and Suggestions for Authors

  • Antidepressants and antianxiety Medication appears several times in the results and discussion but nothing mentioned in the introduction. The authors need to add some information in the introduction to reflect them.

Reply:

Added in page 2

  • What this study adds and recommendations to future work not included.

Reply:

Added in page No. 13

  • Last paragraph in the discussion has details not belong to this study.

Reply:

Removed

  • Objective number 4 has not been achieved in this study. Authors need to consider whether remove it or address it in this study.

Reply:

Removed

  • For interpretation of the results it would be good if the reader had access to the different questionnaires such as HADS and EQ-5 to be able to see the individual questions. Could they be put as an appendix.

Reply:

We will add them, but we don’t prefer to publish them because some of the survey elements belong to other manuscripts

Reviewer 2 Report

·       I think its better to bring the Objectives in the method section

·       Please correct the reference format in the text. (Error! Reference source not found.?)

·       SPSS version..?

·       Method section cab be improved for more clear understanding of the results section.

·       Table 7, please use generic name of medications

·       The results section cab be more clear

Regard, 

Author Response

Reviewer2:

Comments and Suggestions for Authors

  • I think its better to bring the Objectives in the method section

Reply:

Moved to page 2

  • Please correct the reference format in the text. (Error! Reference source not found.?)

Reply:

Fixed

  • SPSS version..?

Reply:

SPSS V.25

  • Method section cab be improved for more clear understanding of the results section.

Reply:

We modified the methods section to be more clear

  • Table 7, please use generic name of medications

Reply:

Name of medications in table 7 provided by the participants. We used their responses on open-ended question where the answer left plank to them since they may not be aware about generic names of these medications. We can replace the names to generic terms, but we believe this study could be beneficial to all people with MS, we prefer to keep these names simple and understandable

  • The results section cab be more clear

Reply:

We reorganized the result to be clearer for the reader

Reviewer 3 Report

Comments are in the file attached.

Author Response

Reviewer3:

This study has aimed to explain the pattern of the medicine administration by people with MS in the UK MS register.

 Line 11- Incomplete sentence

Reply:

Fixed

 Line 16- Grammatical mistake “Which representing”

Reply:

Fixed

 Line 37- In the sentence “joining is open to any person has a”, “who” is missing

Reply:

Fixed

 Line 47- The style of the reference is different than then the rest of the references

Reply:

Fixed

Line 78- The verb in the sentence “to analysis…”, is wrong.

Reply:

Fixed

Line 111- “categorized of” not grammatically correct.

Reply:

Fixed

Line 216- The sentence “Similar frequencies…” does not have a verb/make sense.

Reply:

Fixed

Line 229- Age range should be corrected.

Reply:

Fixed

  • Multiple references are missing.

Reply:

Fixed

  • Amitriptyline in this study is mentioned to be the most efficient in depression management among all other antidepressants. Is that the case? This medication nowadays is less frequently used in the cases of depression due to its low effectivity. Do you think there could be any specific reason that patients with MS are mostly using/prescribed this drug? This could be briefly discussed

Reply:

It was written in line [364] that Cipriani et al., (2018) evidenced Amitriptyline as more efficient in depression management than other antidepressants. In addition, it was written in line [365] that Khurshid et al., (2018) stated Amitriptyline was called the gold-standard antidepressant in that time. Amitriptyline was found used by 2.2% of the cohort of people with MS in this study as mentioned in line [492]. Taking into consideration that 58% of the study population were in the age group (41-60) as shown in table 1, it can be predicted that Amitriptyline was either prescribed to people with MS since long time ago and these people do not like to change. Medication treatment of depressed people should be individualized as per Chang et al., (2022). Or it was prescribed in combination with other medicines to reach a desired upshot of depression management as per Henssler et al., (2022) and Julia et al., (2022).

References:

Chang, F., Kuang, X., & Liu, Y. (2022). The Development and Mechanism of Treatment of Depression. Highlights in Science, Engineering and Technology, 8, 133–142. https://doi.org/10.54097/hset.v8i.1120

Henssler J, Alexander D, Schwarzer G, Bschor T, Baethge C. Combining Antidepressants vs Antidepressant Monotherapy for Treatment of Patients With Acute Depression: A Systematic Review and Meta-analysis. JAMA Psychiatry. 2022;79(4):300–312. doi:10.1001/jamapsychiatry.2021.4313.

Julia A. Koretski, Anthony J. Rothschild, Chapter 24 - Treatment-resistant psychotic depression, Editor(s): Joao Quevedo, Patricio Riva-Posse, William V. Bobo, Managing Treatment-Resistant Depression, Academic Press, 2022, Pages 355-367, ISBN 9780128240670.

  • Besides the grammatical mistakes mentioned here, there are multiple similar mistakes/incomplete sentences throughout the text.

Reply:

We will send the manuscript for language editing once all comments satisfy you

Reviewer 4 Report

In the manuscript titled ‘ Determining current medications usage withing a cohort of patients in the UK. A descriptive retrospective study. The authors did a retrospective analysis of prescribed medicines for multiple sclerosis (MS) in the UK.

Minor comments

1)      Line 11 missing the word UK I believe.

2)      Some of the abbreviation’s full forms are missing and abbreviations came before the full name, for example MS first appeared in line 10 however its full name is given in line 33. Correct other similar formats in the article.

3)      Line 382 there is an extra bracket ).

4)      Table 3, write full name of the country NI=Northern Ireland.

Major comments:

1)     Line 156 Results section: Why there is high gender variation is it because the disease prevalence is more in women or its only because they participated more in the questionnaires?

2)     Table 7: Tysabri, Rebif, and Avonex are brand names, not drugs. Rebif, and Avonex are the same monoclonal antibody (interferon beta-1a). The percent tab and analysis will change, please look into it.

3)     The percentage of Tysabri (natalizumab) and interferon beta-1a is higher than any other medications, but the authors focused more on Amitriptyline any reason for this.

4)     Are there any combination therapy taken by patients that can be included particularly for the disease-modifying agents?

Author Response

Reviewer4:

  • Line 11 missing the word UK I believe.

Fixed

  • Some of the abbreviation’s full forms are missing and abbreviations came before the full name, for example MS first appeared in line 10 however its full name is given in line 33. Correct other similar formats in the article.

Fixed

  • Line 382 there is an extra bracket ).

Fixed

4)      Table 3, write full name of the country NI=Northern Ireland.

Fixed

Major comments:

  • Line 156 Results section: Why there is high gender variation is it because the disease prevalence is more in women or its only because they participated more in the questionnaires?

According to United Kingdom Multiple Sclerosis Register the number of registered females is 16074 and the number of registered males is 6692 which reflect the gender variation in our study

See https://www.ukmsregister.org/Research/OurData  

2)     Table 7: Tysabri, Rebif, and Avonex are brand names, not drugs. Rebif, and Avonex are the same monoclonal antibody (interferon beta-1a). The percent tab and analysis will change, please look into it.

Name of medications in table 7 provided by the participants. We used their responses on open-ended question where the answer left plank to them since they may not be aware about generic names of these medications. We can replace the names to generic terms, but we believe this study could be beneficial to all people with MS, we prefer to keep these names simple and understandable

3)     The percentage of Tysabri (natalizumab) and interferon beta-1a is higher than any other medications, but the authors focused more on Amitriptyline any reason for this.

We focused on Amitriptyline as antidepressant drug to achieve study objective number one.

4)     Are there any combination therapy taken by patients that can be included particularly for the disease-modifying agents?

We didn’t ask the participants about any combination therapy that they take, but this can be mentioned as recommendation for future work.

Round 2

Reviewer 2 Report

Dear Authors,

 Authors responded to my comments. I recommend to accept the manuscript in present form.

Regards,